# Introducing *SoNHR*–Reporting guidelines for Social Networks In Health Research

Douglas A. Luke[1]*, Edward Tsai[2], Bobbi J. Carothers[1], Sara Malone[3], Beth Prusaczyk[4], Todd B. Combs[1], Mia T. Vogel[5], Jennifer Watling Neal[6], Zachary P. Neal[6]

1 Center for Public Health Systems Science, Washington University in St. Louis, St. Louis, MO, United States of America, 2 Office of Community Engagement and Health Equity, University of Illinois Cancer Center, University of Illinois-Chicago, Chicago, IL, United States of America, 3 Department of Surgery, School of Medicine, Washington University in St. Louis, St. Louis, MO, United States of America, 4 Institute for Informatics, Data Science, and Biostatistics, School of Medicine, Washington University in St. Louis, St. Louis, MO, United States of America, 5 Brown School, Washington University in St. Louis, St. Louis, MO, United States of America, 6 Department of Psychology, Michigan State University, East Lansing, MI, United States of America

* dluke@wustl.edu

**Data Availability Statement:** All relevant data are within the paper and its Supporting Information files.

**Funding:** Research reported in this publication was supported by the following sources: -

## Abstract

### Objective

The overall goal of this work is to produce a set of recommendations (*SoNHR*–Social Networks in Health Research) that will improve the reporting and dissemination of social network concepts, methods, data, and analytic results within health sciences research.

### Methods

This study used a modified-Delphi approach for recommendation development consistent with best practices suggested by the EQUATOR health sciences reporting guidelines network. An initial set of 28 reporting recommendations was developed by the author team. A group of 67 (of 147 surveyed) experienced network and health scientists participated in an online feedback survey. They rated the clarity and importance of the individual recommendations, and provided qualitative feedback on the coverage, usability, and dissemination opportunities of the full set of recommendations. After examining the feedback, a final set of 18 recommendations was produced.

### Results

The final SoNHR reporting guidelines are comprised of 18 recommendations organized within five domains: *conceptualization* (how study research questions are linked to network conceptions or theories), *operationalization* (how network science portions of the study are defined and operationalized), *data collection & management* (how network data are collected and managed), *analyses & results* (how network results are analyzed, visualized, and reported), and *ethics & equity* (how network-specific human subjects, equity, and social justice concerns are reported). We also present a set of exemplar published network studies

Implementation Science Centers in Cancer Control (National Cancer Institute and the Barnes Jewish Hospital Foundation; P50CA244431; DAL, ET, SM, BP) - Washington University Institute of Clinical and Translational Sciences (National Center for Advancing Translational Sciences; UL1TR002345; DAL, BJC, TBC) - Centers for Diabetes and Translational Research (National Institute of Diabetes and Digestive and Kidney Diseases, P30DK092950; DAL) - Improving Alzheimer's Disease and Related Dementias Care in Rural Areas (National Institute of Aging; K01AG071749; BP) - T32 Training Program (National Heart, Lung, and Blood Institute; T32 HL130357; MTV) - T32 Training Program (National Cancer Institute; T32CA190194; ET) The funders had no role in study design, data collection and analysis, decision to publish, or preparation of the manuscript.

**Competing interests:** The authors have declared that no competing interests exist.

which can be helpful for seeing how to apply the SoNHR recommendations in research papers. Finally, we discuss how different audiences can use these reporting guidelines.

## Conclusions

These are the first set of formal reporting recommendations of network methods in the health sciences. Consistent with EQUATOR goals, these network reporting recommendations may in time improve the quality, consistency, and replicability of network science across a wide variety of important health research areas.

## Introduction

Despite the dominance of the medical model of health in the 20th and 21st centuries [1], we have always understood that health is very much socially determined. Family and peer influences on smoking and diet; the role of social support on longevity and quality of life; social class and income inequality influencing access to health care; social isolation as a risk factor for depression and suicide; relational and structural factors shaping the course of pandemics– these are just some of the examples of how social factors are involved with individual, community, and population health [2]. Network science is the use of relational and structural theories to study network representations of complex social and physical systems. Although the roots of network science go back over a hundred years, its application within the health sciences is more recent. Driven by theoretical advances [3], modern computational power, and the increased availability of socially structured health data [4], the application of network designs and analytic methods within the health sciences has increased dramatically in the past few decades [5]. For example, Fig 1 shows the increase in network analytic studies over the past 20 years—with health-related network studies currently accounting for as much as 26% of all network publications. Despite this rapid increase in health-science-focused network studies, there are few guidelines for how to best report and disseminate network results in health research. This is particularly important for network studies, as they comprise study designs and analytic approaches that differ dramatically from more traditional health research (e.g., experimental designs based on randomization, or analytic approaches based on the general linear model) [6].

Modern health science has benefited greatly from the reporting guidelines movement, where methodological, analytical, and reporting best practices are used to improve research quality, validity, and impact [7, 8]. Numerous reporting guidelines exist across the social and health sciences that help guide practice and reporting of clinical trials (i.e., CONSORT), systematic reviews (PRISMA), epidemiologic studies (STROBE), and implementation science studies (StaRI), to name just a few. More specifically, broader development and use of reporting guidelines have a critical role to play in ameliorating the reproducibility crisis confronting the health and social sciences [9, 10]. However, until now there has not been a similar set of guidelines for reporting the results of network studies in the health sciences. This is an important gap to fill, most directly because of the prevalence of network studies, as suggested above [11]. However, a set of network reporting best practices will also benefit a notable training gap among network analysis professionals. Training in network science and analytic methods is still not regularly featured in most medical schools, schools of public health, and even many social science departments.

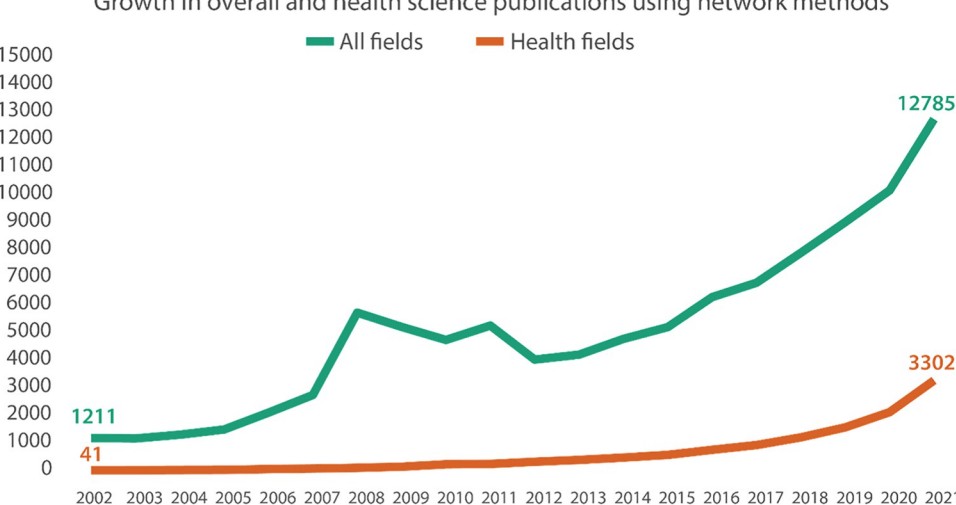

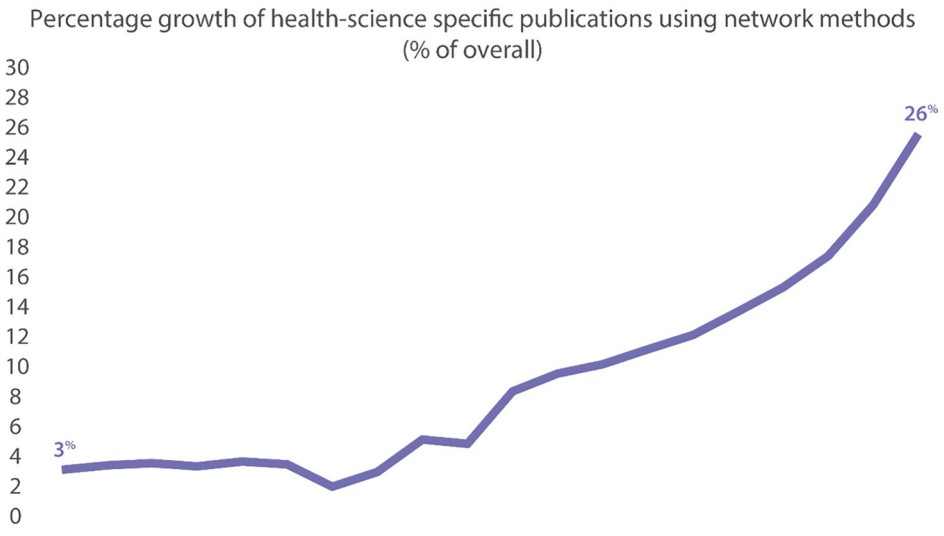

**Fig 1. While network analysis studies have grown in general over the past two decades, the percentage of those specifically from health fields has grown from just 3% in 2002 to 26% in 2021.** (The data for this figure come from SCOPUS searches for 'network analysis' first overall and then limited to publications focusing on health fields).

Although network methods can be used in similar ways across many scientific disciplines, network studies in health research have many distinctive characteristics [12]. Health studies often focus on *social* networks, where the relational systems are made up of social entities such as people or organizations. These kinds of networks exhibit different structural properties (such as closure and transitivity) and are created or formed in different ways compared to networks in the physical or biological sciences. These social networks are implicated in the social and physiological pathways connecting broader social and cultural conditions to downstream health and disease outcomes [2, 3]. Numerous distinctive theories drive health network research, including health communications theory, contagion theory, peer influence, and diffusion of innovations [12–16]. Finally, the rise of network thinking and methods has revolutionized the way we study sexually transmitted diseases such as HIV/AIDS [17], global

pandemics such as COVID [18, 19], and the dissemination and adoption of new health discoveries [20], among many others.

The overall goal of this work is to produce a set of recommendations that will improve the reporting and dissemination of social network concepts, methods, data, and analytic results within health research. These recommendations, called Social Networks in Health Research (SoNHR), focus specifically on *reporting*, not on how to do better network science. There already exist many excellent network analysis texts, for example, those by Scott (2017) [21] or Borgatti, Everett, and Johnson (2018) [22]. And although the primary emphasis is on reporting network studies in peer-reviewed scientific publications, we anticipate that these recommendations will be useful in other contexts including evaluation reporting, and in training and teaching. These reporting guidelines will be disseminated broadly to network science and health research audiences, including via registration with the EQUATOR (Enhancing the QUAlity and Transparency Of health Research; https://www.equator-network.org/) health research guidelines network [23]. This paper is structured in three parts: first, we describe the process we took for developing the network reporting guidelines. Second, we present and discuss the recommendations themselves. We then conclude with some recommendations on how to use the recommendations in different contexts and present some helpful resources including examples of the recommendations in practice.

## Methods

### Overview

This is a guidelines development study with the main goal of producing a set of reporting recommendations for studies using network data and methods in the health sciences. The general methodological approach we used for developing these recommendations was a modified expert panel consensus process, as recommended by EQUATOR and implemented by other reporting guidelines development teams [24]. This study was approved as exempt research by the Institutional Review Board of Washington University in August 2021 (ID: 202108053). Participants provided consent online prior to survey completion as part of the recruitment process.

### Development of initial set of recommendations and expert panel survey

An initial set of network reporting recommendations was developed by the author team, based on an informal review of the network science and health science literatures, as well as their experience using network methods. More specifically, we started this process by conducting rapid reviews [25] of network applications in health research, as well as network study reporting guidelines. A rapid review process was justified here, because the goals of this guideline development project did not require a more formal systematic review, and a rapid review could confirm our hypothesis that there were few if any existing health research network guidelines [26]. While we found extensive use of health-related network methods (see Fig 1), we quickly found that there were no existing published studies focused on network reporting recommendations. (See S4 File for more technical information, including database details and the literature search keywords.) Collectively, the author team has decades of experience conducting network studies, disseminating network studies to health and social science audiences, teaching and training students in network methods, publishing network methods texts [27] and reviewing grants and manuscripts using network and systems science methods. Members of the team have also served as editors of network science journals, and advised federal agencies on network and systems science methods in public health (including CDC, NCI, NHLBI, USDA, and the National Academy of Sciences). In this development phase, we produced a set

of 28 candidate recommendations, grouped into five categories: 1) conceptualization, 2) operationalization, 3) data collection & management, 4) analyses & results, and 5) ethics & equity. These categories broadly represent the phases of empirical research, along with an overarching category focusing on ethical and equity aspects of network science [28].

Originally, we planned to convene network science experts to provide input and feedback on the recommendations. However, because of COVID-19 we switched to an online survey approach. The survey was developed in Qualtrics and focused on two types of feedback. Usability ratings were assessed using two items: *importance* of the recommendation, and *clarity* of the recommendation. Importance was defined as "the degree to which you feel that the particular guideline is critical for researchers to understand and follow in their empirical studies which apply SNA methods" and was rated on a 1 (Not at all Important) to 5 (Very Important) scale. Clarity was defined as "the degree to which the wording of the guideline clearly communicates to researchers using these reporting guidelines the type of information needed to satisfy the guideline" and was rated on a 1 (Not at all Clear) to 5 (Very Clear) Likert scale.

In addition to these quantitative items, participants were asked to respond to a series of open-ended questions focusing on general improvements to the guidelines, suggestions for disseminating the guidelines, and how they might use the guidelines in their own work. More specifically, one of the open-ended feedback questions was "What is the single most important improvement you would like to see in the above draft SNA reporting guidelines?" S1 File contains the expert panel feedback survey, including the initial set of 28 recommendations.

## Participants

Our goal was to identify individuals who were experts in social network analysis applied to health contexts. To that end, we identified five stakeholder groups that should be represented: experts in applied health research, social network methodologists, network methods instructors, journal editors, and funders. Using these groups as guiding principles, we identified individuals who authored well-known papers, books, and analysis software; colleagues from federal funding agencies (NIH, CDC, etc.); participants in network analysis trainings (e.g., Systems Science for Social Impact); and members of the International Network for Social Network Analysis (INSNA) as well as Women in Network Science (WINS). This approach reflected our desire to ensure broad participation coverage, including strong participation by women and international network scientists. We invited a total of 147 people to participate, of whom 67 responded for a response rate of 45.6%. The survey ran from December 2021 through April 2022. Responses were treated as confidential; specific identifying information was stripped from the data files prior to analyses.

Our response rate was slightly above average for online surveys [29], and particularly strong for a survey with no specific monetary incentives. Despite limited information on non-respondents, we were able to compare participants to non-respondents on a couple characteristics. Women were more likely to participate (63% vs. 45%; Chi-square = 3.90, $p$ = .048). There were more participants from the U.S., but this difference was not significant (85% vs. 70%, Chi-square = 3.67, $p$ = .055).

Participant characteristics are presented in Tables 1 and 2. The participants as a group were experienced network and health scientists, working across a wide variety of fields and research areas. Over 80% of the participants had at least six years of experience using network designs and methods, with over 20% having at least two decades of experience. Nearly all participants had published network papers and reviewed network-focused papers or grants during the past five years (94% and 88%, respectively). The participant group included ten members who have worked in an organization that funds network research activities. These characteristics suggest

**Table 1. Professional characteristics of expert panel members.** (Percentages can add up to more than 100% because of multiple choice options).

|  | Frequency | Percentage |
|---|---|---|
| *Network Research Experience* |  |  |
| 20+ years | 14 | 21.5 |
| 11–20 years | 22 | 33.8 |
| 6–10 years | 18 | 27.7 |
| 1–5 years | 11 | 16.9 |
| *Network Science Activities (past 5 years)* |  |  |
| Published paper | 61 | 93.8 |
| SNA professional organization | 58 | 89.2 |
| Reviewed SNA papers/grants | 57 | 87.7 |
| Taught SNA | 48 | 73.8 |
| Worked in org. funding SNA | 10 | 15.4 |
| *Primary Disciplinary Focus* |  |  |
| Public health | 38 | 58.5 |
| Sociology | 29 | 44.6 |
| Statistics, Math, CS | 13 | 20.0 |
| Organizational research | 13 | 20.0 |
| Psychology | 11 | 16.9 |
| Medicine | 8 | 12.3 |
| Social Work | 7 | 10.8 |
| All others (15) | 31 | 47.6 |
| *Health Science Applications* |  |  |
| Implementation science | 25 | 38.5 |
| Health disparities, equity, justice | 25 | 38.5 |
| Community-focused | 22 | 33.8 |
| Methods development | 21 | 32.3 |
| Program evaluation | 19 | 29.2 |
| Health services | 18 | 27.7 |
| Mental health | 17 | 26.2 |
| Epidemiology | 16 | 24.6 |
| Health promotion | 15 | 23.1 |
| Addiction, substance use | 14 | 21.5 |
| Health policy | 13 | 20.0 |
| Children, youth, families | 13 | 20.0 |
| Older adults | 12 | 18.5 |
| Cancer | 11 | 16.9 |
| Infectious disease | 11 | 16.9 |
| All others (17) | 67 | 101.5 |

that we were successful at recruiting a large, diverse, and experienced group of network and health researchers who could adequately inform development of the network reporting guidelines.

## Development of final set of network reporting recommendations

Data from the expert panel survey were summarized and analyzed. Specifically, descriptive statistics and distributional characteristics of the importance and clarity items were examined to identify recommendations that could be dropped or recommendations that needed

**Table 2. Demographic characteristics of expert panel members.**

|  | Frequency | Percentage |
|---|---|---|
| *Gender Identity* |  |  |
| Woman | 41 | 63.1 |
| Man | 23 | 35.4 |
| Prefer not to say | 1 | 1.5 |
| *Identify as Transgender* |  |  |
| No | 64 | 98.5 |
| Prefer not to say | 1 | 1.5 |
| *Race* |  |  |
| White | 53 | 81.5 |
| Asian | 7 | 10.8 |
| Black or African American | 1 | 1.5 |
| Other race not listed | 2 | 3.1 |
| Prefer not to say | 4 | 6.2 |
| *Hispanic/Latinx Ethnicity* |  |  |
| No | 59 | 90.8 |
| Yes | 6 | 9.2 |
| *Current Home Institution* |  |  |
| Domestic (U.S.) | 56 | 84.8 |
| International | 10 | 15.2 |

improvement. Thematic summaries were prepared for the open-ended questions and were also used to help differentiate between strong and weaker recommendations.

## Results

### Recommendations development

The quantitative results of the preliminary network reporting recommendations from the Delphi survey are presented in Table 3. In general, the preliminary recommendations received high scores on *importance* and *clarity*. The operationalization recommendations scored the highest, with average importance ratings of 4.7 (out of 5) and clarity ratings of 4.6. The five preliminary visualization recommendations scored lowest in importance (mean = 3.5). Closer examination of the visualization ratings suggested that participants fell into one of two categories—some felt these recommendations were quite important, but a substantial number clearly felt that visualizations in general were less important in network science. The detailed, item-level survey responses (including dot-plots) are available in S1 Table and S1 Fig.

**Table 3. Summary of importance and clarity ratings of preliminary set of network reporting recommendations.**

| Domain | No. of Recs. | Importance | | Clarity | |
|---|---|---|---|---|---|
|  |  | *Mean* | *Range* | *Mean* | *Range* |
| Conceptualization | 3 | 4.3 | 4.1–4.6 | 4.2 | 3.7–4.4 |
| Operationalization | 4 | 4.7 | 4.4–4.9 | 4.6 | 4.4–4.7 |
| Data Collection & Management | 6 | 4.3 | 4.0–4.6 | 4.1 | 3.8–4.5 |
| Analysis & Results—Description | 2 | 4.3 | 4.2–4.4 | 4.1 | 4.0–4.2 |
| Analysis & Results—Visualization | 5 | 3.5 | 3.1–3.8 | 4.1 | 3.9–4.3 |
| Analysis & Results–Modeling & Simulation | 4 | 4.4 | 4.2–4.7 | 4.2 | 3.9–4.4 |
| Ethics & Equity | 4 | 4.2 | 3.8–4.6 | 4.3 | 3.8–4.5 |

The responses to the open-ended questions indicated general enthusiasm for the network reporting recommendations (e.g., "These are awesome and well needed. Bravo!"), but more importantly included many suggestions on how to improve individual recommendations as well as their overall structure. There were 56 separate responses to the open-ended question asking participants to list the single-most important improvement they would like to see made to the recommendations. Many (23%) of the recommendations were about improving the wording of specific items (N = 13), as well as a number of suggestions (N = 11, 20%) for adding new items. Most of the recommendations were actually about how the guidelines should be presented in this paper (N = 14, 25%). There were only a few suggestions for adding a new domain (N = 1, 2%), dropping a specific item (N = 1, 2%), or dropping an entire domain (N = 3, 5%). Participants also provided numerous suggestions on how to disseminate the guidelines as well as how they might use the recommendations in their own work (see Discussion below).

Taking the importance, clarity, and qualitative feedback into account, the team revised the recommendations by dropping, rewriting, and reordering items. Items were dropped if they received lower importance or clarity ratings, or if on further reflection they seemed to be redundant with other items or were too narrow in scope. For example, one of the preliminary 28 guideline recommendations that was dropped was, "For valued networks, describe reconciliation of conflicting values when provided by both members of the dyad." This item was overly complicated (receiving the lowest clarity score in its group) and it applies to relatively few types of network studies. The main structural change we made was to combine two separate recommendation sections (*Network Description* and *Network Visualization*) into one section. This last modification was partly driven by a number of participants who suggested in the qualitative feedback part of the survey that we did not need so many recommendations specific to network visualization.

## Final reporting guidelines for Social Networks In Health Research (SoNHR)

Table 4 presents our final set of 18 recommendations for reporting social network methods and results, particularly within health sciences research (SoNHR). These recommendations are not a formal checklist; we do not expect all these recommendations to apply to every specific network paper. However, the use of these recommendations will help ensure that network methods and results are reported as clearly as possible, benefiting future research.

**Conceptualization.** The first two recommendations refer to the basic *conceptualization* of the research project as reported in the publication. Network studies are inherently structural or relational; they concern themselves with how social objects (e.g., persons, organizations) relate to one another, or how social structures shape the flow of physical or social things such as viruses, information, money, behavior, etc. [15]. The first recommendation is to make the structural and relational conceptualizations that drive the research questions clear to the reader. This provides the basic rationale for using network methods in the first place. The second recommendation is similar but focuses on the network methods themselves. How are these network methods used in the study necessary or advantageous, given the research questions driving the study?

**Operationalization.** The second set of recommendations is focused on *operational definitions* of the various pieces of the network study, *i.e.*, who is in the network, what the ties represent, and what kind of network is being analyzed. Definitions of the nodes are usually straightforward, but they may need additional details if the nodes are not persons (e.g., organizations), or in the case of 2-mode networks, where both types of nodes need to be clearly

**Table 4. Final reporting guidelines—Social Networks in Health Research (SoNHR).**

| Conceptualization |
| --- |
| C1. Clearly describe how and why networks are relevant for addressing the study's research question(s). |
| C2. Make the value of a network analysis apparent by explaining what kind of information a network analysis tells us that a more traditional approach would not. |
| **Operationalization** |
| O1. Define the *nodes* to make it clear what a node represents. |
| O2. Define the *ties* so that it is clear what each type of tie represents. Make sure to indicate if ties are directed or undirected and if ties are binary or weighted. |
| O3. Define the boundaries of the network so that it is clear who is included and not included in the network. |
| O4. State clearly the basic type of network that is being analyzed (*e.g.*, complete network, ego networks, affiliation/ 2-mode/bipartite networks). |
| **Data Collection & Management** |
| D1. Describe network data collection *procedures* and tools (e.g., surveys and software) in enough detail to support replication. When possible, provide access to all surveys, instruments, and tools used. |
| D2. Describe the network *data* used in the study, including pre-existing data sources, how the data are stored, managed, and whether/where they are publicly available. |
| D3. Discuss missingness in the network data, its implications, and any attempts to impute or account for missing data (*e.g.*, rationale for requiring one or both responses when only one member of a dyad reports a relationship). |
| D4. Report all data transformations (*e.g.*, aggregation of person-level to organization-level nodes, reconciliation of conflicting link values when provided by both members of a dyad, etc.). |
| **Analyses & Results** |
| *Description & Visualization* |
| AD1. When discussing network statistics, be clear about the unit of analysis (*e.g.*, node, dyad, sub-network, whole network). |
| AD2. Report network statistics (*e.g.*, centrality, centralization, homophily, etc.) in terms of the real-world property of the setting being assessed (*e.g.*, what does it mean for a particular node to have high degree centrality, for a network to have high betweenness centralization, or for nodes with similar characteristics to cluster together?) |
| AD3. Network visualizations should clearly illustrate study findings using design principle best practices that are appropriate for network characteristics and the goals of the visualization (*e.g.*, using node color or shape to convey categorical properties, using node size to convey quantitative properties, limited use of labels and different shapes in large networks, varying line weights or colors in small networks). |
| *Modeling & Simulation* |
| AM1. Explain the theoretical foundations that drive the model or simulation development and testing. |
| AM2. If statistical network models or simulations are used, clearly specify model mechanisms and outcomes (e.g., tie formation). When possible, provide reproducible statistical programming code used in the analyses to support replication. |
| AM3. Present some information on how well the network model fits with the observed network data and discuss any important implications of model fit. |
| **Ethics & Equity** |
| E1. Discuss how confidentiality was explained to participants and how their confidentiality was ensured, including considerations of identifiability in network visualizations and reporting. Clarify how participants understood that information could be collected about them even if they chose not to participate, *or* that non-participation precluded information being collected about them. |
| E2. Discuss any potential biases within network structures and results that may be rooted in the network study methods (*e.g.*, failure to capture complete networks, organizational or specific group non-participation, over-representation). Think in terms of equity and social, economic, and health justice when doing so. |

defined. For example, in an infectious disease contact tracing study, one type of node would be people who are infected, and the other type of node would be defined as a specific physical location where people come into contact with one another [30].

Defining the network ties used in a study often requires careful attention to detail. At a minimum, the description of network ties should include a statement of what the tie represents (e.g., contact frequency, trust, sexual contact) and how it was measured (e.g., self-report using

a 5-point scale with each anchor point defined). This is particularly important when more than one type of tie is collected and analyzed in the study. Moreover, once ties have been defined, it is a good practice to refer to that type of tie specifically (e.g., 'friendship tie') rather than a more generic term like 'ties,' or 'connections.' Otherwise, this can lead to the impression by the reader that a network has only one type of relationship among network members when any network contains many different kinds of social relationships, whether measured in the study or not.

It is always important to describe who is in the network, and, conversely, who is not in the network. Thus, the boundaries of the network should be clear, especially for complete network studies [31]. And this leads to the last recommendation in this section, which is a clear description of the type of network being analyzed. First, state whether this network is *complete*, where most or all members of a boundary-defined network are included (e.g., 'all cancer healthcare providers in St. Louis County); or *ego-centric*, where networks are constructed from the perspective of individual members or 'egos' (e.g., personal support networks of people in a study of the effects of social support). In addition, it is often important to define the network in terms of the types of ties—are the ties directed or non-directed, and are the ties valued or binary?

**Data collection & management.**　The third set of recommendations concerns reporting of *data collection and management* details. Network data are quite different from the data used in the vast majority of other health and social science research. They are fundamentally relational, which implies different kinds of data collection, management, and subsequent analytic practices. Therefore, authors may need to provide more information on these data management steps than they normally would. Taken together, these data management recommendations support future replication. They also suggest network-specific data issues. For example, missing data in network studies represent more serious threats to accurate interpretation and bias [32, 33], so it is important to describe how missing data are handled in any network study. Furthermore, a common data issue in survey-based network analysis is when individuals in a dyad give conflicting information about a shared tie. If this is the case, then authors should report how that conflicting information is resolved [34]. In addition, enough details on any network data transformation (e.g., turning 2-mode into 1-mode network data [30]) should be presented to support full understanding and potential future replication.

**Analyses & results.**　The first set of recommendations under analyses and results helps ensure that readers will fully understand the results coming out of basic *descriptive network analysis* studies. First, make sure to clarify what the network *unit-of-analysis* is for any particular set of network statistics. Readers may not fully appreciate how network results can focus on individual network members, pairs of network members, subnetworks, or complete network characteristics. Second, it is very important to convey how a particular network statistic measures or captures an underlying behavioral, structural, or relational characteristic of the network or network member. For example, in a communication network study, do not simply report the technical formula for a statistic such as network *centralization*, but describe how this measure assesses the extent to which the communication structure is more or less hierarchical in nature. Finally, many network studies employ visualizations, so it is important to design them in such a way that it is clear how the visualization supports or reveals the underlying network characteristic or analytic result. This is likely to require both general information visualization skills [35], as well as network-specific graphic design principles [36, 37].

The second section of recommendations under analyses and results refers to good practices for reporting results from *network modeling or simulation* studies. These types of studies move beyond simple network descriptions and visualizations to using data and theory to pose and test hypotheses about network structures and influences. Taken together, these three

recommendations will help ensure that readers will understand the theory or framework that drives the network modeling analyses, will have enough technical information to support further research replication using these models, and appreciate how well the network model or simulation fits with the observed network data [38].

**Ethics & equity.** The final set of recommendations refers to special considerations around *ethics and equity* in network studies. Standard non-network surveys and similar methods of data collection allow for anonymity and confidentiality. However, when collecting data on networks anonymity is not possible, as the individuals or organizations in the network must be named to create ties in subsequent analyses. Even if a person or organization chooses not to participate in a network study, they might be named by others during data collection. Offering informed consent or even *truly informed consent* for participants in network studies is a promising practice to ensure confidentiality [28, 39]. As part of this process, those choosing not to participate should be aware that they may be named by others. Omitting non-consenters in a network study is dangerous and can greatly impact network structures, analyses, results, and interpretations [40]. More appropriate strategies for maintaining confidentiality are to anonymize data as soon as possible after data collection and to reduce the number of people who have access to the data [39, 41].

Equity is an essential consideration in all scientific research, and in network analysis its importance is amplified. As stated above, maximizing response rates and capturing complete networks are also especially important in network studies. It is important to identify all who influence or are affected by the substantive area of the network under consideration, including historically underrepresented types of network members such as patients in individual networks or relatively small agencies in organizational networks. Researchers should monitor recruitment and look for patterns in non-responses. For example, one study found that network studies of sex workers commonly underrepresented men in their roles as sex workers and designed a purposeful sampling strategy more inclusive of them [42]. Similar to the ethical concerns around reporting incomplete network study findings, the underrepresentation of certain groups or organizations can distort results, or even worse, exacerbate existing inequities through exclusion.

## Discussion

The SoNHR (Social Networks in Health Research) guidelines were developed to promote the clear and comprehensive reporting of network studies in the literature among health researchers. Already a well-established theoretical and methodological frame, the use of network science in health research continues to increase. However, without reporting guidelines, the utility, replicability, and impact of network science studies are diminished. These guidelines were created through an iterative, expert consensus building process and the resulting recommendations can be applied broadly across health research studies, including those in the clinical, social science, community, and population/public health research fields.

### Suggested audiences and supporting resources

These guidelines have the potential to benefit at least five distinct audiences–instructors, authors, journal editors, reviewers, and readers–that capture the full process of scientific knowledge generation and dissemination.

First, these guidelines offer *authors* recommendations about how to report the design and results of their health-related social network research clearly and consistently. We anticipate this is especially helpful and important because social network methods are both diverse and rapidly developing, which can lead to confusion among authors about what essential aspects to

report. Although the guidelines themselves offer specific recommendations, Table 5 provides examples of published health sciences research that have followed these recommendations, and thus that can serve as exemplars for authors. Although not exhaustive, these examples represent a collection of "best practices" for each of the recommendations and illustrate how each of the recommendations can be put into action.

Second, these guidelines can be used by *instructors* of social network coursework to inform the topics covered in such courses, and by their students to rapidly develop an understanding of the most important features of social network research. For example, modules in an introductory course on social network analysis could roughly follow the major headings of the guidelines, while individual recommendations under each of these headings could serve as learning objectives within these modules.

Third, these guidelines can serve as a helpful checklist for *reviewers* when evaluating health-related social network research. Reviewers can use the guidelines and individual recommendations to ensure the completeness of their evaluations of social network studies. In addition, following these guidelines to evaluate social network research may improve the consistency between reviewers, which is often quite low [76], as they provide a common set of reporting standards to evaluate.

Fourth, these guidelines can be required as reporting standards by *journal editors* who publish health-related social network research to ensure the standardized reporting of social network methods and results. This type of editorial requirement is not without precedent. For example, many journal editors already require reporting guidelines developed for other methods including PRISMA for systematic reviews and meta-analyses [77], JARS-QUAL for qualitative research [78], JARS-QUANT for quantitative research [79], and CONSORT for randomized trials [80].

Finally, these guidelines have benefits for *readers* of health-related social network research. When implemented, these guidelines should improve the detail, clarity, and transparency of social network methods and results for readers. Additionally, when put into practice, these guidelines should ensure that critical methodological and analytic details are provided in social network papers, making it easier to re-use published social network research in meta-analyses and systematic reviews.

## Benefits to the health sciences

Social networks are situated within a social-ecological context that allows us to understand (and study) health and disease outcomes as products of socially-mediated structures and processes [3]. This highlights the importance of network methods and theories to the health sciences. Adoption of the SoNHR reporting guidelines can benefit the health sciences in at least three broad ways, by enhancing the *validity* of network health studies, by better integrating aspects of *equity* into health science, and by improving the *clarity and consistency* of network reporting. By improving the validity, equity, and clarity of these studies, the overall quality, quantity, and impact of network-related health research may also be increased [8, 81]. Table 5 provides examples of health sciences research that realized these benefits by using reporting methods consistent with the SoNHR guidelines.

First, a major theme of these recommendations is strengthening the connection between theory, methods, and results when using network analysis (see particularly recommendations C1, C2, O1, O2, AD2, and AM1), which all work to enhance the construct validity of the study data and results. That is, the recommendations always point to a tight tie between specific network statistics and metrics, and their conceptual or theoretical underpinnings. For example, our understanding of the relational dimensions of health is strengthened by studies that

**Table 5. Examples of published health science studies demonstrating network reporting recommendations.**

| Rec. | 1st Author & Year | Title | Description | Pointer |
|---|---|---|---|---|
| *Conceptualization* | | | | |
| C1 | Bearman (2004) [43] | Chains of affection: The structure of adolescent romantic and sexual networks. | Links theoretical models of disease transmission to network properties. | Pages 47–52 |
| C1 | DeLay (2017) [44] | Assessing the impact of homophobic name calling on early adolescent mental health: A longitudinal social network analysis of competing peer influence effects. | Provides a theoretical rationale for the relational nature of homophobic name calling and for studying network selection and influence processes. | Page 957 |
| C2 | McGlashan (2019) [45] | Collaboration in complex systems: Multilevel network analysis for community-based obesity prevention Interventions. | Lays out the role of multilevel network analysis for unpacking the complexities inherent in community-based systems interventions for obesity prevention. | Abstract |
| C2 | Valente (2019) [46] | Effects of a social-network method for group assignment strategies on peer-led tobacco prevention programs in schools. | Links the social science literature on peer influence to the rationale for the network intervention. | Page 1837, last paragraph |
| *Operationalization* | | | | |
| O1 | Lee (2011) [47] | Social network analysis of patient sharing among hospitals in Orange County, California. | Clearly describes the nodes as hospitals and providers characteristics of these hospitals. | Last paragraph of Background and first paragraph of Methods. |
| O1 | Fuller (2007) [48] | Use of social network analysis to describe service links for farmers' mental health. | The nodes are described as "agencies known to have contact with" farming families about mental health and wellbeing in a specific town. | Second paragraph in Methods. |
| O2 | Koehly (2003) [49] | A social network analysis of communication about hereditary nonpolyposis colorectal cancer genetic testing and family functioning. | Ties were clearly defined through measures of communication, cohesiveness, affective involvement, leadership, and conflict. | Measures sub-section under "Materials and Methods" section. |
| O2 | Bruening (2018) [50] | Friendship as a social mechanism influencing body mass index (BMI) among emerging adults. | The description effectively communicates that ties represent friendships and are binary and directed. | Page 4 in "Friendships" subsection of Measurements |
| O3 | McGlashan (2019) [45] | Collaboration in complex systems: Multilevel network analysis for community-based obesity prevention Interventions. | Clear operational definition of network boundaries, specifically how network members were formally engaged in the steering committees. | Participants subsection within the Methods section. |
| O3 | Cauchemez (2010) [19] | Role of social networks in shaping disease transmission during a community outbreak of 2009 H1N1 pandemic influenza. | Clearly describes inclusion criteria based on an outbreak investigation in Pennsylvania. | 1st paragraph in Results section. |
| O4 | Broccatelli (2021) [51] | Social network research contribution to evaluating process in a feasibility study of a peer-led and school-based sexual health intervention. | Defined variables of interest as the offline friendship ties among students and student's embeddedness in online Facebook groups. | Second paragraph of Analytical Method section. |
| O4 | Dhand (2019) [52] | Social networks and risk of delayed hospital arrival after acute stroke. | Links network metrics (e.g., size and connectivity) to the underlying scientific constructs (e.g., conduits for information entry, exchange, and disruption). | Last paragraph of Introduction. |
| *Data Collection & Management* | | | | |
| D1 | Van der Gaag (2005) [53] | The Resource Generator: social capital quantification with concrete items. | Specific resource generator items and a clear definition about what counts as "knowing" a person are provided. | Table 2. |
| D1 | Lorant (2017) [54] | Optimal network for patients with severe mental illness: a social network analysis. | Survey questions and response choices are clearly explained. | Two paragraphs under Data Collection sub-section |
| D2 | Kim (2020) [55] | COVID-19 health communication networks on Twitter: Identifying sources, disseminators, and brokers. | Description of a pre-existing, publicly-available dataset and its existing structures is clear. Provides a good demonstration of how to sample the most important nodes from a large dataset. Discussion of how retweets and mentions are used to measure indegree and outdegree and highlight information sources. | Pages 131–132 "Construction of retweet and mention networks" section. Page 133 "User level" section. |
| D2 | Ahmed (2020) [56] | COVID-19 and the 5G conspiracy theory: Social network analysis of Twitter data. | Twitter dataset is described clearly, including what hashtags were mined | Two paragraphs under Methods section |

(*Continued*)

**Table 5.** (Continued)

| Rec. | 1st Author & Year | Title | Description | Pointer |
|------|-------------------|-------|-------------|---------|
| *Conceptualization* | | | | |
| D3 | Gile (2017) [57] | Analysis of networks with missing data with application to the National Longitudinal Study of Adolescent Health. | Demonstrates ERGMs of a network with missing data using four approaches: all observations (as-is), complete-case (removing nodes of non-responders), incomplete-case (uses all nodes but not the links between any dyads including non-responders), and differential popularity (models differences between respondents and non-respondents to partially compensate). | Pages 507–517, "Analyzing adolescent friendship networks" section. |
| D3 | De Moor (2020) [58] | Assessing the missing data problem in criminal network analysis using forensic DNA data. | Information on how previous research has handled missing data and how the current study handled their missing data is clearly presented | Section 4.2 under the Methods section |
| D4 | Carothers (2022) [59] | Mapping the lay of the land: Using interactive network analytic tools for collaboration in rural cancer prevention and control. | Discusses aggregation of individual responses to agency-level nodes, symmetrization of non-directed networks, and reconciliation of one valued network and one directed network. | Pages 1160–1161, "Network data management" section. |
| D4 | Scott (2005) [60] | Social network analysis as an analytic tool for interaction patterns in primary care practices. | Describes how direct observations were transcribed into directed and undirected adjacency matrices that then formed the basis of the network data. | Pages 444–445 |
| *Analyses & Results (Description & Visualization)* | | | | |
| AD1 | de la Haye (2017) [61] | The dual role of friendship and antipathy relations in the marginalization of overweight children in their peer networks: The TRAILS Study. | Clearly indicated the unit of analysis in ERGMS: "The unit of the analysis is the ordered pair of students in a classroom (xij), and the dependent variable is the observed value of a friendship or dislike tie (1 = present, 0 = absent)." | Page 3 |
| AD1 | Hawe (2008) [62] | Use of social network analysis to map the social relationships of staff and teachers at school. | Analyses are clearly described in terms of node vs. network unit if analysis, such as centralization. | Paragraphs 5–10 under Methods section. |
| AD2 | Weeden (2021) [63] | Still a small world? University course enrollment networks before and during the COVID-19 pandemic. | Interpreted changes in node degree, proportion of students in largest bi-component, average distance, and within-field clustering within a university course enrollment network before and during the COVID-19 pandemic. | p. 76–79 |
| AD2 | Cappella (2013) [64] | Classroom peer relationships and behavioral engagement in elementary school: The role of social network equity. | Introduces, defines, and interprets a new network statistic measuring social network equity. | Page 371 |
| AD3 | Block (2020) [18] | Social network-based distancing strategies to flatten the COVID-19 curve in a post-lockdown world. | Good use of figure design (especially node color and tie bolding) to clearly communicate shortest pathways and infection risk. | Figs 1 & 2 |
| AD3 | Ahn (2011) [65] | Flavor network and the principles of food pairing. | Good use of color, node size, and tie size to designate food groups and network structures. | Fig 2 |
| *Analyses & Results (Modeling & Simulation)* | | | | |
| AM1 | Bond (2014) [66] | Friends or foes? Relational dissonance and adolescent psychological wellbeing. | Provides a description of ERGMs & the foundations underlying ERGM models. Fig 1 provides a description of what each parameter included in the model tests. | Page 4 & Fig 1 |
| AM1 | Kiuru (2012) [67] | Is depression contagious? A test of alternative peer socialization mechanisms of depressive symptoms in adolescent peer networks. | Tests two different mechanisms of socialization (contagion & convergence) and describes how different parameters (average alter & average similarity) were used in SAOMs to test each. | Page 252 |
| AM2 | Combs (2022) [68] | Simulating the role of knowledge brokers in policy making in state agencies: An agent-based model. | "A week in the life of an agent" illustrates how opinions about health policies diffuse across the social networks in the agent-based model. | Sections 2.3.4 |

**Table 5.** (Continued)

| Rec. | 1st Author & Year | Title | Description | Pointer |
|------|-------------------|-------|-------------|---------|
| *Conceptualization* | | | | |
| AM2 | Mercken (2009) [69] | Dynamics of adolescent friendship networks and smoking behavior: Social network analyses in six European countries. | Clearly lays out how the network model is able to distinguish between social influence and social selection mechanisms for adolescent smoking behavior. | Fig 1 & Table 2 |
| AM3 | Luke (2010) [70] | Systems analysis of collaboration in 5 national tobacco control networks. | Presents model fit statistics and graphics for the main ERGM model. | Fig 2 and text on page 1294 |
| AM3 | Morris (2009) [71] | Concurrent partnerships and HIV prevalence disparities by race: linking science and public health practice. | Conducts a simulation to examine how assortative mixing drives racial disparities in HIV prevalence; compares results of the simulations to observed survey data, particularly AddHealth. | Results section, particularly the 2nd column on page 1027 |
| *Ethics & Equity* | | | | |
| E1 | Maya-Jariego (2021) [72] | Confidentiality, power relations and evaluation of potential harm in the study of the personal and organizational networks of travel agents in Moscow | Part 1.4 describes ethics in the study of the networked Moscow tourism industry. | Pages 57–58 |
| E1 | Banbury (2017) [73] | Can videoconferencing affect older people's engagement and perception of their social support in long-term conditions management: a social network analysis from the Telehealth Literacy Project. | 'Procedure' section discusses how researchers explained confidentiality to participants and obtained informed consent. | Page 940 |
| E2 | Browne (2005) [74] | Snowball sampling: using social networks to research non-heterosexual women. | Entire paper examines the equity implications of using snowball sampling for studying 'hidden' populations in network studies. | Whole paper |
| E2 | Lobb (2014) [75] | Using organizational network analysis to plan cancer screening programs for vulnerable populations. | The 'Discussion' section explains that although community health centers are important in cancer care networks, they are often overlooked or presented as peripheral in network studies. | Pages 361–363 |

explicitly link real-world phenomena such as disease transmission [19] or psychological well-being [66] to specific network properties or metrics.

Second, these reporting guidelines will help to ensure important aspects of participant equity and research ethics are clearly communicated (see particularly O3, E1, and E2). For example, because the richness of network data can more easily re-identify participants, which is particularly risky when also collecting sensitive health information, it is essential to ensure that participants' understand these risks and how their confidentiality will be protected [73].

Finally, these recommendations will help to ensure the clarity and consistency of health sciences research. As Table 5 illustrates, the use of network methods in health sciences research is becoming increasingly common. However, attempts to replicate or meta-analyze network analyses of health phenomena have previously identified a lack of clarity as a key barrier. For example, one systematic review of network effects on youth eating behaviors noted that "neither outcomes nor summaries of social network characteristics were sufficiently homogenous to undertake statistical meta-analysis" [82]. Similarly, another review focused on psychosis noted that "we aimed to perform meta-analysis but this was not possible due to the heterogeneity of social network definitions" [83].

## Strengths, gaps, and next steps

This is among the first set of reporting guidelines for social network science concepts, methods, and results. Although a number of network *data* formatting and reporting guidelines exist [84, 85], to our knowledge there are no guidelines focusing on best reporting practices in scientific dissemination, especially within health research. These guidelines are being published

in EQUATOR, which will increase their visibility and longevity and hopefully enhance their effectiveness. Finally, these reporting guidelines include equity and ethical considerations, which are important, require special consideration in network science, and have previously been underdiscussed.

Although we developed these reporting guidelines to be broadly applicable to many kinds of network studies, they may be less helpful (or need to be adapted) for some kinds of network study designs or analytic frameworks (e.g., purely qualitative network studies, ego-centric designs, dynamic analyses, etc.). Similarly, the scope of these recommendations is somewhat high-level, we deliberately left out extremely specific reporting suggestions (e.g., guidance around the use of weighted ties in network visualizations) that would not be applicable to many network studies. Finally, for the expert panel consensus process, we may have missed some important viewpoints and distinct experiences that could have helped inform the network reporting guidelines. However, the relatively large, experienced, and diverse set of participants we ended up with (see Tables 1 & 2) somewhat mitigates that concern.

We will continue to work on disseminating the SoNHR guidelines beyond EQUATOR and then hosting these guidelines on relevant websites (including at cphss.wustl.edu). We also encourage interested partners to further spread the guidelines and share how they have used them (see above). In particular, case studies of the application of the network reporting recommendations would be useful for the field of network health research.

As we gain a better understanding of how social connections, structures, and dynamics are implicated in human disease processes, effective healthcare delivery, and promotion of public health, there will be a concomitant need for appropriate scientific frameworks, study designs, and analytic approaches. Network science and social network analysis are particularly well-suited for these types of health research. The network reporting guidelines presented here will be helpful for ensuring that the knowledge generated in these studies can have the broadest impact, both scientifically and socially.

## Supporting information

**S1 File. Appendix A.**
(DOCX)

**S2 File. Feedback data.**
(CSV)

**S3 File. Feedback data variable dictionary.**
(CSV)

**S4 File. Appendix D.**
(DOCX)

**S1 Table. Appendix B.**
(DOCX)

**S1 Fig. Appendix C.**
(DOCX)

## Acknowledgments

We are grateful for the contributions and expertise of our recommendations development expert panel. They were crucial for helping us develop and finalize the SoNHR recommendations.

## Author Contributions

**Conceptualization:** Douglas A. Luke, Edward Tsai, Beth Prusaczyk, Todd B. Combs.

**Data curation:** Bobbi J. Carothers.

**Formal analysis:** Bobbi J. Carothers.

**Funding acquisition:** Douglas A. Luke, Beth Prusaczyk.

**Methodology:** Douglas A. Luke, Edward Tsai, Bobbi J. Carothers, Beth Prusaczyk, Mia T. Vogel, Jennifer Watling Neal, Zachary P. Neal.

**Project administration:** Douglas A. Luke.

**Visualization:** Todd B. Combs.

**Writing – original draft:** Douglas A. Luke, Edward Tsai, Sara Malone, Todd B. Combs, Jennifer Watling Neal, Zachary P. Neal.

**Writing – review & editing:** Douglas A. Luke, Edward Tsai, Bobbi J. Carothers, Sara Malone, Beth Prusaczyk, Todd B. Combs, Mia T. Vogel, Jennifer Watling Neal, Zachary P. Neal.

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
