## [Decision Letter · Decision Letter 0]

4 Jul 2023

PONE-D-23-11338Introducing *SoNHR – Reporting guidelines for social networks in health research**PLOS ONE*

Dear Dr. Luke,

Thank you for submitting your manuscript to PLOS ONE. After careful consideration, we feel that it has merit but does not fully meet PLOS ONE’s publication criteria as it currently stands. Therefore, we invite you to submit a revised version of the manuscript that addresses the points raised during the review process.

In particular, the reviewers have requested a more detailed description and justification of the methodological approach, and suggested to strengthen the section dedicated to the guidelines/recommendations. 

*A rebuttal letter that responds to each point raised by the academic editor and reviewer(s). You should upload this letter as a separate file labeled 'Response to Reviewers'.*

*A marked-up copy of your manuscript that highlights changes made to the original version. You should upload this as a separate file labeled 'Revised Manuscript with Track Changes'.*

*An unmarked version of your revised paper without tracked changes. You should upload this as a separate file labeled 'Manuscript'.*

**

*We look forward to receiving your revised manuscript.*

*Kind regards,*

*Stefano Ghinoi, Ph.D.*

Academic Editor

*PLOS ONE*

**

Reviewers' comments:

*Reviewer's Responses to Questions*

*

**Comments to the Author**
*

1. Is the manuscript technically sound, and do the data support the conclusions?

*The manuscript must describe a technically sound piece of scientific research with data that supports the conclusions. Experiments must have been conducted rigorously, with appropriate controls, replication, and sample sizes. The conclusions must be drawn appropriately based on the data presented. *

*Reviewer #1: Partly*

*Reviewer #2: Partly*

*Reviewer #3: Partly*

*2. Has the statistical analysis been performed appropriately and rigorously? *

*Reviewer #1: Yes*

*Reviewer #2: Yes*

*Reviewer #3: N/A*

*3. Have the authors made all data underlying the findings in their manuscript fully available?*

*The PLOS Data policy requires authors to make all data underlying the findings described in their manuscript fully available without restriction, with rare exception (please refer to the Data Availability Statement in the manuscript PDF file). The data should be provided as part of the manuscript or its supporting information, or deposited to a public repository. For example, in addition to summary statistics, the data points behind means, medians and variance measures should be available. If there are restrictions on publicly sharing data—e.g. participant privacy or use of data from a third party—those must be specified.*

*Reviewer #1: Yes*

*Reviewer #2: Yes*

*Reviewer #3: No*

*4. Is the manuscript presented in an intelligible fashion and written in standard English?*

*PLOS ONE does not copyedit accepted manuscripts, so the language in submitted articles must be clear, correct, and unambiguous. Any typographical or grammatical errors should be corrected at revision, so please note any specific errors here.*

*Reviewer #1: Yes*

*Reviewer #2: Yes*

*Reviewer #3: Yes*

*5. Review Comments to the Author*

*Please use the space provided to explain your answers to the questions above. You may also include additional comments for the author, including concerns about dual publication, research ethics, or publication ethics. (Please upload your review as an attachment if it exceeds 20,000 characters)*

*Reviewer #1: Dear Authors,*

Thank you for the opportunity to review this paper. I found it relevant, timely and well written. Unfortunately, I also think that the current version of your paper suffers from some limitations. I have a few comments that hopefully will help you improve the paper. They are arranged by section.

INTRODUCTION

I think the motivation of this study could be strengthened and explained better. I appreciate that your idea is to draw a parallel between the benefits of reporting guidelines for other aspects of health science and the need for similar guidelines for network studies. Although I find this rationale meaningful, I would encourage you to contextualize your motivation more explicitly, explaining why reporting guidelines are needed for network science and what the risks/implications are of ignoring this need.

Furthermore, the category of "network studies in health sciences" is potentially very broad. Do you plan to develop a set of guidelines that suit any type of network studies/level of analysis/research questions? I think it would be worth clarifying this upfront as well as providing examples of what studies you have in mind and/or mentioning sub-streams of literature.

METHODS

1. I am not fully convinced by the approach you adopt to draft the initial 28 recommendations. While I do not doubt your extensive experience in the field (and the advantages of drawing on this), I believe your paper would benefit from the adoption of a more robust approach. This would also make the readers more confident about the conclusions you draw and ensure this initial list is not biased. This is relevant because – to the best of my understanding (pp. 10 and 12)– you asked the experts to provide suggestions on how to improve the listed recommendations, but not to indicate additional ones.

You could conduct a systematic – instead of an informal – literature review and arrange the findings into a table, where you list the most recurrent recommendations (setting a reasonable threshold for “recurrent”) that you can derive from extant literature. Then, you could match them with your own experience – briefly explaining and justifying the recommendations you have derived from your own work.

2. You report that the response rate to your expert panel survey is 45.6 %. I would encourage you to comment on this response rate, specifically on non-respondents. Did you check that the sample of respondents is not biased? Does it cover all key characteristics that you used to identify the initial sample?

DISCUSSION

It seems to me that most of the guidelines you provide apply to network studies across a variety of fields, not to health sciences specifically. Given the focus of your study, I encourage you to discuss in more detail – and in a more compelling way – the reasons why your guidelines are beneficial to health sciences, which drawbacks of existing studies they would allow addressing, which implications they would have for the reliability of results and policy implications. I appreciate your attempt to convey this message by explaining which audiences would benefit from your study, but I would recommend that you expand this section providing examples of research questions, empirical settings, etc. This consideration resonates with my comment on the introduction – and I think addressing both comments would strengthen your contribution.

*Best of luck with the next steps.*

*Reviewer #2: This paper provides a set of 18 recommendations called as SoNHR- Social Networks in Health Research focusing on reporting and dissemination of social networks analysis within health research community. The research identifies the key gap of not having guidelines that help researchers report the study results of using social networks analysis in healthcare setting.*

The paper addresses a relevant domain of study; however, at this point, the research is at a preliminary stage. Minor revisions are needed, especially on the following points:

- The initial set of recommendations that the authors provide are based on an “informal review”. The authors need to clarify what this exactly means and provide concrete measures to showcase how this review was conducted. The authors need to clarify what “their experience using network methods is” and substantiating this claim to show how exactly this is relevant in the set of recommendations provided.

- More empirical analysis are needed in the first stage of the set of recommendation;

*- Results are well presented. However, the set of 18 recommendations are quite generic at the moment. The recommendations can benefit from concrete examples specifically contextualised to the setting.*

*Reviewer #3: Thanks for giving me the opportunity to read and review the paper. I found it interesting for the potential implications that are described, as it highlights the relevance of social sciences - and SNA more in particular - in health care studies. Unfortunately, I was not captured by any original contribution the paper should have produced. Recommendations are presented as very broad, not focused on the specific setting under investigation, and I did not find any original idea on how SNA methods may be adapted to health care. This is originally what I expected to find in the manuscript. Even if I appreciated very much the efforts of cross fertilization, again, I did not find an extact streamline the paper would contribute to.*

*6. PLOS authors have the option to publish the peer review history of their article (what does this mean?). If published, this will include your full peer review and any attached files.*

**

*Reviewer #1: No*

*Reviewer #2: **Yes: **Anna Piazza*

*Reviewer #3: No*

**

*While revising your submission, please upload your figure files to the Preflight Analysis and Conversion Engine (PACE) digital diagnostic tool, https://pacev2.apexcovantage.com/. PACE helps ensure that figures meet PLOS requirements. To use PACE, you must first register as a user. Registration is free. Then, login and navigate to the UPLOAD tab, where you will find detailed instructions on how to use the tool. If you encounter any issues or have any questions when using PACE, please email PLOS at figures@plos.org. Please note that Supporting Information files do not need this step.*

---

## [Author Response · Author response to Decision Letter 0]

25 Sep 2023

See attached file that contains our detailed responses.

---

## [Decision Letter · Decision Letter 1]

14 Nov 2023

PONE-D-23-11338R1Introducing SoNHR – Reporting guidelines for social networks in health researchPLOS ONE

Dear Dr. Luke,

Thank you for submitting your manuscript to PLOS ONE. After careful consideration, I feel that you have been able to address almost all the comments from the reviewers: however, while one reviewer has suggested to accept your paper, the second one still requires some minor revisions (in the Method section). Therefore, I invite you to submit a revised version of the manuscript that addresses the points raised during the review process.

We look forward to receiving your revised manuscript.

Kind regards,

Stefano Ghinoi, Ph.D.

Academic Editor

PLOS ONE

Journal Requirements:

Reviewers' comments:

Reviewer's Responses to Questions

**Comments to the Author**

1. If the authors have adequately addressed your comments raised in a previous round of review and you feel that this manuscript is now acceptable for publication, you may indicate that here to bypass the “Comments to the Author” section, enter your conflict of interest statement in the “Confidential to Editor” section, and submit your "Accept" recommendation.

Reviewer #1: All comments have been addressed

Reviewer #2: (No Response)

2. Is the manuscript technically sound, and do the data support the conclusions?

Reviewer #1: Yes

Reviewer #2: Yes

3. Has the statistical analysis been performed appropriately and rigorously? 

Reviewer #1: N/A

Reviewer #2: (No Response)

4. Have the authors made all data underlying the findings in their manuscript fully available?

Reviewer #1: Yes

Reviewer #2: Yes

5. Is the manuscript presented in an intelligible fashion and written in standard English?

Reviewer #1: Yes

Reviewer #2: Yes

6. Review Comments to the Author

Reviewer #1: Dear Authors, congratulations on a much improved paper. This revision has addressed all my concerns, henceforth I do not have further comments.

Reviewer #2: Method section: the authors should better explain how they conduct "a rapid review", why this is justifiable (pg.7); and the use of only one database such as Scopus (figure 1). Also, they should clarify the keywords used to conduct this rapid review.

7. PLOS authors have the option to publish the peer review history of their article (what does this mean?). If published, this will include your full peer review and any attached files.

Reviewer #1: No

Reviewer #2: No

---

## [Author Response · Author response to Decision Letter 1]

17 Nov 2023

We have addressed the remaining minor concerns of one of the reviewers. Namely, we have added some additional information on the rapid review process, added a new citation supporting using a rapid review, and added a new supplemental materials file (Appendix D) with technical details about the Scopus review.

---

## [Editor Report · Decision Letter 2]

29 Nov 2023

Introducing *SoNHR – Reporting guidelines for social networks in health research*

*PONE-D-23-11338R2*

*Dear Dr. Luke,*

*We’re pleased to inform you that your manuscript has been judged scientifically suitable for publication and will be formally accepted for publication once it meets all outstanding technical requirements.*

*Within one week, you’ll receive an e-mail detailing the required amendments. When these have been addressed, you’ll receive a formal acceptance letter and your manuscript will be scheduled for publication.*

*An invoice for payment will follow shortly after the formal acceptance. To ensure an efficient process, please log into Editorial Manager at http://www.editorialmanager.com/pone/, click the 'Update My Information' link at the top of the page, and double check that your user information is up-to-date. If you have any billing related questions, please contact our Author Billing department directly at authorbilling@plos.org.*

*If your institution or institutions have a press office, please notify them about your upcoming paper to help maximize its impact. If they’ll be preparing press materials, please inform our press team as soon as possible -- no later than 48 hours after receiving the formal acceptance. Your manuscript will remain under strict press embargo until 2 pm Eastern Time on the date of publication. For more information, please contact onepress@plos.org.*

*Kind regards,*

*Stefano Ghinoi, Ph.D.*

Academic Editor

*PLOS ONE*

* *

*Additional Editor Comments (optional):*

* *
---

## [Editor Report · Acceptance letter]

5 Dec 2023

PONE-D-23-11338R2 

Introducing *SoNHR* – Reporting guidelines for social networks in health research 

Dear Dr. Luke:

I'm pleased to inform you that your manuscript has been deemed suitable for publication in PLOS ONE. Congratulations! Your manuscript is now with our production department. 

Kind regards, 

on behalf of

Dr. Stefano Ghinoi 

Academic Editor

PLOS ONE